# Platycodonis Radix Alleviates LPS-Induced Lung Inflammation through Modulation of TRPA1 Channels

**DOI:** 10.3390/molecules28135213

**Published:** 2023-07-05

**Authors:** Tan Yang, Shuang Zhao, Yu Yuan, Xiaotong Zhao, Fanjie Bu, Zhiyuan Zhang, Qianqian Li, Yaxin Li, Zilu Wei, Xiuyan Sun, Yanqing Zhang, Junbo Xie

**Affiliations:** 1College of Traditional Chinese Pharmacy, Tianjin University of Traditional Chinese Medicine, Tianjin 301617, China; yangtan0912@163.com (T.Y.); zs1446193347@163.com (S.Z.); yuan19980727@163.com (Y.Y.); zhang-zy19@tjutcm.edu.cn (Z.Z.); lqq961016@163.com (Q.L.); weizilu35@163.com (Z.W.); sunxy@tjutcm.edu.cn (X.S.); 2Department of Chemistry, Cleveland State University, Cleveland, OH 44115, USA; x.zhao20@vikes.csuohio.edu; 3College of Biotechnology and Food Science, Tianjin University of Commerce, Tianjin 300134, China; bufanjie00@163.com (F.B.); zhyqing@tjcu.edu.cn (Y.Z.); 4Department of Pathology and Laboratory Medicine, Weill Cornell Medicine, New York, NY 10065, USA; yal4011@med.cornell.edu

**Keywords:** platycodonis radix, platycodins, TRPA1, lung inflammation

## Abstract

Platycodonis Radix (PR), a widely consumed herbal food, and its bioactive constituents, platycodins, have therapeutic potential for lung inflammation. Transient Receptor Potential Ankyrin 1 (TRPA1), which is essential for the control of inflammation, may be involved in the development of inflammation in the lungs. The aim of this study was to determine the TRPA1-targeted effects of PR against pulmonary inflammation and to investigate the affinity of PR constituents for TRPA1 and their potential mechanisms of action. Using a C57BL/6J mouse lipopolysaccharides (LPS) intratracheal instillation pneumonia model and advanced analytical techniques (UPLC-Q-TOF-MS/MS, molecular docking, immuno-fluorescence), five platycodins were isolated from PR, and the interaction between these platycodins and hTRPA1 was verified. Additionally, we analyzed the impact of platycodins on LPS-induced TRPA1 expression and calcium influx in BEAS-2B cells. The results indicated that PR treatment significantly reduced the severity of LPS-triggered inflammation in the mouse model. Interestingly, there was a mild increase in the expression of TRPA1 caused by PR in healthy mice. Among five isolated platycodins identified in the PR extract, Platycodin D_3_ (PD_3_) showed the highest affinity for hTRPA1. The interaction between platycodins and TRPA1 was verified through molecular docking methods, highlighting the significance of the S5–S6 pore-forming loop in TRPA1 and the unique structural attributes of platycodins. Furthermore, PD_3_ significantly reduced LPS-induced TRPA1 expression and calcium ion influx in BEAS-2B cells, substantiating its own role as an effective TRPA1 modulator. In conclusion, PR and platycodins, especially PD_3_, show promise as potential lung inflammation therapeutics. Further research should explore the precise mechanisms by which platycodins modulate TRPA1 and their broader therapeutic potential.

## 1. Introduction

Transient Receptor Potential Ankyrin 1 (TRPA1), first identified in human lung fibroblasts in 1999, is a non-selective calcium ion channel pivotal to the normal functioning of the respiratory system [1]. It is widely distributed in the lung and airway, including lung fibroblasts, alveolar epithelial cells, and primary sensory neurons [2]. Various exogenous and endogenous stimuli, such as PM 2.5, cigarette smoke, reactive oxygen species, and inflammatory cytokines, play a vital role in inciting respiratory inflammation by regulating the TRPA1 channel [3,4]. Besides increasing the expression of TRPA1, these agonists can also open the channel, instigating a calcium influx, and promoting the release of SP, CGRP, and inflammatory cytokines, thereby exacerbating the inflammatory process [5]. Consequently, TRPA1 has emerged as a key target for treating respiratory tract inflammation, particularly lung inflammation, making the search for novel TRPA1 channel modulators essential for the development of safe and effective drugs for pneumonia [6].

The consumption of herbal products is increasing in many regions [7]. It is estimated that 80% of the world’s population relies on herbal medicines [8]. Most of the synthetic drugs currently used for pain relief and analgesia have many side effects and toxicity [9]. Plants remain a vast, untapped source of structurally novel compounds that could serve as lead compounds for new drugs. Platycodonis Radix (PR), the root of *Platycodon grandioflorum* (Jacq.) A.DC., is prevalent in Eastern Asia and has been employed as a renowned medicine food homology species for the prevention and treatment of various lung diseases [10]. Platycodins are considered the principal bioactive components of PR [11]. Recently, platycodin D has been shown to significantly attenuate cigarette smoke-induced lung pathological changes and the infiltration of inflammatory cells [12,13]. However, the mechanisms by which Platycodonis regulates inflammation remain elusive.

The purpose of this study was to determine the TRPA1-targeted effects of PR against pulmonary inflammation and to investigate the affinity of PR for TRPA1 and its underlying mechanism of action. In this study, the TRPA1 receptor-mediated effects of PR against lung inflammation were investigated in lipopolysaccharides (LPS)-induced pneumonia mice. Furthermore, the primary potentially active ingredients were screened and identified using 2D ultra-performance liquid chromatography–quadrupole-time-of-flight tandem mass spectrometry (UPLC-Q-TOF-MS/MS). The binding free energy and binding mode of platycodin ligands with TRPA1 were investigated via molecular docking. Finally, to confirm its role in modulating the TRPA1 channel, the effects of PD_3_ on TRPA1 expression levels and calcium ion influx in BEAS-2B cells were assessed. Our results show for the first time that PR alleviates lung inflammation by regulating TRPA1. Building upon these findings, future research could further explore the impact of other platycodins on TRPA1 modulation and, subsequently, their role in mitigating lung inflammation. Understanding these interactions could potentially pave the way for the development of novel therapeutic strategies for respiratory tract inflammation.

## 2. Results

### 2.1. PR Alleviates LPS-Induced Lung Inflammation through Reducing TRPA1 Expression in Mice

To delineate the impact of PR on LPS-induced lung inflammation, we established a robust experimental model (Figure 1A). As shown in Figure 1B, the results of the histopathologic examination of the lungs showed that the LPS-exposed group exhibited distinct signs of pneumonia, such as thickened alveolar walls, extensive hemorrhage, and fibrin deposition, as opposed to the control group. DEX pre-treatment significantly improved the LPS-induced abnormal histopathological changes in mice. Interestingly, PR treatment mitigated these deleterious effects on the lung tissue in a dose-responsive manner. TNF-α and IL-1β are two key pro-inflammatory cytokines, the increased concentration of which often indicates the severity of pneumonia and the intensity of the inflammatory response. As demonstrated in Figure 1C, the ELISA results measuring the concentrations of TNF-α and IL-1β in the BALF revealed a significant increase in the production and release of these cytokines in LPS-induced pneumonia, leading to a heightened inflammatory response in the lung tissues. However, the administration of PR was shown to suppress or reduce the production and release of TNF-α and IL-1β in a dose-responsive manner, which decreased their concentrations in the BALF, thereby reducing the inflammatory response and preserving the lung tissue. This observation is consistent with the results of the histopathologic examination of the lungs. Collectively, these findings suggest that PR may be an effective therapeutic strategy for the treatment of pneumonia.

TRPA1, known as a nociceptor, plays a key role in the development of tissue inflammation [14]. It has been demonstrated that LPS can trigger rapid, membrane-delimited excitatory responses via TRPA1. However, it is important to note that the direct effects of LPS might not be solely attributed to TRPA1. Prior studies showed that Toll-like receptor 4 (TLR4) and TRPV1 may also contribute to the direct effects of LPS [15]. To further explore the role of TRPA1 in LPS-induced pneumonia, its activity was manipulated using an inhibitor (A967079) and an activator (AITC) (Figure 1D). The results revealed a significant decrease in LPS-triggered inflammatory factors (TNF-α and IL-1β) in the BALF upon A967079 treatment, while AITC preconditioning amplified this response (Figure 1E). Moreover, LPS treatment markedly increased TRPA1 expression in the mouse lung at both mRNA and protein levels, an effect that was mitigated by A967079 and enhanced by AITC (Figure 1F,G). These results highlight the crucial role of TRPA1 in LPS-induced lung damage.

Finally, to validate whether PR exerts its anti-inflammatory effect via TRPA1 modulation, the mRNA and protein expression levels of TRPA1 were assessed (Figure 2). As depicted in Figure 2B,C, PR attenuated the LPS-stimulated mRNA and protein expression of TRPA1 in a dose-dependent manner. This effect was enhanced when A967079 was co-administered with PR, leading to a further decrease in the mRNA and protein expression of TRPA1. Conversely, the application of AITC was observed to increase the mRNA and protein expression of TRPA1. However, in the (0.8, 1.6 mg/kg) PR-AITC-LPS group, the inhibitory effect of both concentrations appeared to be similar. In addition, at the same concentrations (0.8, 1.6 mg/kg), the PR-AITC-LPS and PR-LPS groups appeared to undergo similar inhibitory effects, but still showed significant differences compared to the LPS group.

Taken together, these findings indicated that PR could potentially alleviate LPS-induced lung inflammation by reducing TRPA1 expression.

### 2.2. PR Exhibits a Certain Elevating Effect on TRPA1 Expression in Normal Mice

The aforementioned findings confirmed that PR mitigated LPS-induced lung inflammation by downregulating TRPA1 expression in mice. However, the influence of PR on the TRPA1 levels in normal mice remained unclear. To address this, normal mice were i.g. administered PR, followed by the measurement of the mRNA and protein expression levels of TRPA1 (Figure 2D–F). Compared to the control group, neither low nor medium doses of PR induced significant alterations in the mRNA and protein expression levels of TRPA1. Interestingly, a high dose of PR demonstrated a mild up-regulatory effect on the mRNA and protein expression levels of TRPA1, although this effect was notably weaker than that induced by LPS. 

Collectively, these findings suggested that PR indeed has the capacity to modulate TRPA1 expression, leading to potential health implications.

### 2.3. Five Primary Platycodins Identified in a PR Water Extract 

Understanding the composition of natural compounds like PR is of paramount importance, as it provides insights into their medicinal attributes and sheds light on the mechanisms through which they exert therapeutic effects. The 2D UPLC-Q-TOF-MS/MS analytical technique is recognized for its high resolution and sensitivity in component analysis, making it an ideal tool for scrutinizing the composition of PR extracts.

Triterpene saponins, the most abundant component in PR, were also identified as major components [16]. The main cleavage pathway of triterpene saponins involves sequential intramolecular deglycosylation and interchain cleavage to produce characteristic fragment ions. Taking PD_3_ as an example, based on the mass spectrogram results obtained from 2D UPLC-Q-TOF-MS/MS, a peak was observed at 8.768 min (Figure 3A). The glycosidic bond was broken under a high-energy bombardment. In negative ion mode, an [M+HCOO]^−^ ion at *m*/*z* 1431.6670 was seen to generate fragments at *m*/*z* 1385.6387 [M–H]^−^, 1253.6019 [M-H-C_5_H_8_O_4_]^−^, and 843.4632 [M-H-C_21_H_34_O_16_]^−^ (Figure 3B). The peak was tentatively identified as PD_3_ by comparison of the retention time and mass spectrometric ion fragmentation with those of standards and further validated by reference to the characteristic mass fragmentation pattern reported in the available literature [17,18]. Other identified platycodins were list in Table 1. In summary, five main platycodins components were identified in PR extracts: PD_3_, Platycodin D, Platycodin D_2_, Deapio-Platycodin D, and Platycodgenin. These results offer a clearer picture of the potential therapeutic components present in PR.

### 2.4. Platycodins Interaction with TRPA1: Molecular Docking

Molecular docking is the study of how two or more molecular structures fit together, to predict how a protein (enzyme) interacts with small molecules (ligands). To further elucidate the mechanism of action of the complex consisting of TRPA1 and platycodins, their affinity and binding mode were evaluated through molecular docking (Figure 4). The results demonstrated that the affinity values (binding free energy) for the best conformations of the five platycodins were −7.700 kcal·mol^−1^ (PD_3_), −7.460 kcal·mol^−1^ (Platycodigenin), −7.448 kcal·mol^−1^ (Platycodin D), −7.196 kcal·mol^−1^ (Platycodin D_2_), and −7.007 kcal·mol^−1^ (Deapio Platycodin D). Here, the negative sign signifies a spontaneous reaction. Ligands can spontaneously bind to the active pocket of TRPA1 without needing to absorb external energy. The higher the affinity, the more stable the complexes formed. Among these five compounds, PD_3_ showed the strongest affinity.

The TRPA1 protein comprises 1119 amino acids that assemble into homotetramers. Each subunit includes six transmembrane regions and forms a pore-like structure between the hydrophilic regions of the S5 and the S6 transmembrane domains. The S5 and S6 regions are key sites for agonist and antagonist binding [19]. All these five compounds can be well accommodated within the TRPA1 (S5, S6, and the first pore helix) structure. Acting as molecular wedges, these compounds can inhibit or activate the TRPA1 channel by influencing the movement of its subunits. Interestingly, we found that the Ser 943 residue occurs simultaneously in the binding sites of both AITC–TRPA1 and platycodins–TRPA1 complexes, aligning with the conclusion that residues in the transmembrane core contribute to the electrophilic sensitivity of TRPA1 [20]. This might partly explain why high doses of PR can activate TRPA1 channels in normal mice. Furthermore, we found that the ILE803 residue is the common active site of platycodins and LPS in TRPA1. Combined with in vivo experiments, we speculate that platycodins may exert competitive inhibitory effects on LPS binding sites, thereby alleviating LPS-induced pulmonary inflammation. Conversely, when PR and A967079 were used in combination, a new pharmacological site was created in the TRPA1 binding pocket. This did not interfere with the antagonistic effect of either on TRPA1. This hypothesis is also supported by the structure of the double antagonist of TRPA1.

Platycodins is a triterpenoid compound. Despite numerous studies showing that natural products with various structural features can regulate TRPA1, there have been relatively few reports on triterpenoid compounds [21]. In the HEK293 cell line stably expressing human TRPA1, saikosaponin inhibited the calcium flux stimulated by polydialdehyde, indicating that oleanane triterpenoids have significant potential as TRPA1 antagonists [22]. Platycodigenin demonstrated stronger TRPA1 binding power compared to the other analyzed compounds, which suggests that its specific structure (2β, 3β, 16α, 23, 24-pentahydroxy-olean-12-en-28-oic acid) is key to TRPA1 binding. In the docking model of platycodins binding to TRPA1, the triterpenoid mainly interacts with TRPA1 through van der Waals forces, carbon–hydrogen bonds, and conventional hydrogen bonds. Notably, the platycodigenin-type PD_3_ forms van der Waals bonds with six residues, namely, ASN845, TYR849, TYR812, GLU981, GLN851, and SER985. This interaction may facilitate the molecular fitting and reduce the binding energy. The composition of the sugar chains at positions 3 and 28 significantly impacts the binding activity, particularly the apiose residue of the sugar chain linked at position 28 [23].

### 2.5. PD_3_ Downregulates LPS-Induced TRPA1 Expression and Ca^2+^ Influx in BEAS-2B Cells

Based on the analysis of the results above, PD_3_ displayed the highest affinity for TRPA1. To investigate the ability of PD_3_ to regulate TRPA1, a standard sample of PD_3_ was used to treat BEAS-2B cells, and immunofluorescence staining was utilized to analyze the resulting effects. As shown in Figure 5A, after LPS stimulation, TRPA1 was dispersed throughout each BEAS-2B cell, evidenced by the diffusion of green fluorescence. This indicated a significant increase in TRPA1 expression. However, upon administration of PD_3_, the intensity of green fluorescence in the BEAS-2B cells decreased, suggesting that PD_3_ dose-dependently reduced the level of TRPA1 expression induced by LPS.

TRPA1 is an activated Ca^2+^-permeable cation channel that plays a vital role in the sensitivity to various stimuli [24]. To evaluate the regulatory role of TRPA1, we examined the Ca^2+^ influx under LPS induction. The results indicated that the Ca^2+^ levels were the highest under LPS induction. Nevertheless, when PD_3_ was introduced, the Ca^2+^ levels gradually decreased in correlation with the increasing PD_3_ concentration (Figure 5B). In conclusion, these results suggest that PD_3_ has the potential to downregulate LPS-induced TRPA1 expression and the associated Ca^2+^ influx.

## 3. Discussion and Conclusions

In this study, it was found that PR attenuated LPS-induced lung inflammation by reducing TRPA1 expression, while also having a minor activating effect on TRPA1 expression in the lungs of normal mice, suggesting that PR contains a regulator of TRPA1. Further analysis of potentially active ingredients identified the affinity and binding energy sequence for the five platycodins contained in PR with TRPA1: PD_3_ > Platycodigenin > Platycodin D > Platycodin D_2_ > Deapio Platycodin D. PD_3_, which displayed the strongest binding affinity, was selected for in vitro cell validation. Its significant regulation of the expression of TRPA1 and Ca^2+^ influx suggests PD_3_ as a potential novel channel modulator for TRPA1.

TRPA1 is known to be prevalent throughout the respiratory system and plays a crucial role in the development and progression of inflammation [25]. PR attenuates inflammation by inhibiting iNOS and several pro-inflammatory cytokines [26]. Such findings are consistent with the observed results, suggesting the role of TRPA1 in mediating pulmonary inflammation. In the presence of LPS-induced lung inflammation, PR not only attenuated the inflammation but also improved various inflammatory indicators such as cytokines levels and histopathology. When combined with TRPA1 modulators, PR appeared to alleviate lung inflammation by regulating the TRPA1 receptor. Thus, our in vivo experiments strongly suggest that PR could reduce LPS-induced lung inflammation by decreasing TRPA1 expression.

As one of the main chemical components of PR, platycodins have been considered as the main active substances of PR [27]. Platycodins have been proven to have anti-inflammatory activity [28]. This is consistent with our findings. The results of molecular docking showed that platycodins have different affinity for TRPA1. The specific structure of platycodins may be the primary reason for their anti-inflammatory activity. This finding is also supported by a previous report [29]. The ILE803 residue, a common active site for platycodins and TRPA1, may be essential for their binding. These results suggest that the specific structure of platycodins and an active site in TRPA1 are key factors for the anti-inflammatory effects of these platycodins.

PD_3_, the active ingredient with the highest affinity for TRPA1, was shown to have anti-inflammatory properties. According to our in vitro results, PD_3_ effectively inhibited LPS-induced TRPA1 channel activation and Ca^2+^ influx. Ca^2+^ is an important ion for both activation and modulation of TRPA1 channel activity and a key regulator for potentiation and inhibition of TRPA1 channel activity upon chemical stimulation. PD3 inhibits TRPA1 by inhibiting Ca^2+^ inward flow [30]. Therefore, PD_3_, as a potential novel channel modulator, could alleviate LPS-induced lung inflammation by modulating TRPA1.

These findings suggest that TRPA1 could serve as a novel therapeutic target for PR against pneumonia, and PD_3_ might act as a new channel modulator of TRPA1. Thus, PR could be a potential candidate for the treatment of pneumonia and may provide new possibilities for the prevention and treatment of respiratory diseases.

## 4. Materials and Methods

### 4.1. Reagents and Materials

The original PR medicinal material was sourced from Zhangshu Tianqitang Co., Ltd. (Jiangxi, China), and PR (100 g) was thrice extracted with 1000 mL of distilled water by heating reflux for 2 h. After concentration and filtration, the supernatants were evaporated to yield a PR extract. LPS was obtained from Shanghai Yuanye Bio-Technology Co., Ltd. (Shanghai, China) AITC (TRPA1 agonist) and A967079 (TRPA1 antagonist) were supplied by Merck (Shanghai, China). The Mouse TNF-α High-Sensitivity ELISA Kit and Mouse IL-1β High-Sensitivity ELISA Kit were supplied by Multi Sciences (LIANKE) Biotech, Co., Ltd. (Hangzhou, China). Platycodin D_3_, Platycodigenin, Platycodin D, Platycodin D_2_, and Deapio Platycodin D were provided by Beijing Rongcheng Xinde Technology Development Co., Ltd. (Beijing, China). All other chemicals were of analytical grade. 

### 4.2. Animals and Animal Grouping

C57BL/6J mice (8-week-old, male) were procured from Beijing HFK Science Co., Ltd. (Beijing, China). All animal experiments were authorized by the Tianjin University of Traditional Chinese Medicine animal ethics committee (No: TCM-LAEC2021024). The mice were acclimatized under a 12 h light/dark cycle at a constant temperature (22 ± 2 °C) and were provided with standard chow and water ad libitum. The dosages of PR administered to the mice were based on human doses and the results of probe trials. 

To investigate the alleviating effect of PR on LPS-induced lung inflammation and explore the role of the TRPA1 receptor in pneumonia onset, the mice were randomly divided into various groups (10 mice/group) including control, LPS, dexamethasone (DEX, 6 mg·kg^−1^) (positive control), three PR-LPS (each receiving varying concentrations of PR extract), A967079, AITC, A967079-LPS, and AITC-LPS groups. In each group, 7 mice were selected for lung tissue collection, while the remaining 3 mice were used for the collection of bronchoalveolar lavage fluid (BALF). Specifically, for the PR effect investigation, both the control and the LPS group were given 1 mL of saline intragastrically (i.g.) for 21 days. Following this period, the control and LPS groups received an intratracheal instillation of 100 μL of either saline or LPS (6 mg·kg^−1^), respectively. The PR-LPS groups were i.g. administered the PR extract daily for 21 days, with doses of 0.8, 1.6, and 3.2 mg·kg^−1^, respectively. After this period, each PR-LPS group received an intratracheal instillation of 100 μL of LPS (6 mg·kg^−1^). Lung tissue and BALF were collected from each group 24 h later. In a concurrent experiment exploring the role of the TRPA1 receptor in pneumonia onset, both the control and the LPS group directly received an intratracheal instillation of 100 μL of either saline or LPS (6 mg·kg^−1^), respectively. Lung tissue and BALF were collected from each group 24 h later. The A967079 and AITC groups were administered either A967079 (60 mg·kg^−1^) or AITC (5 mg·kg^−1^) intraperitoneally, respectively. Lung tissue and BALF were collected from each group 1 h later. Lastly, the A967079-LPS and AITC-LPS groups were administered either A967079 (60 mg·kg^−1^) or AITC (5 mg·kg^−1^) intraperitoneally, respectively. After 1 h, both groups received an intratracheal instillation of 100 μL of LPS (6 mg·kg^−1^). Lung tissue and BALF were collected from each group 24 h later.

To assess the influence of PR on lung inflammation via TRPA1 expression modulation and study the effect of PR on TRPA1 expression in normal mice, the mice were randomly divided into various groups (10 mice/group) including control, LPS, three PR-A967079-LPS (each receiving varying concentrations of PR extract), three PR-AITC-LPS (each receiving varying concentrations of PR extract), and three PR groups (each receiving varying concentrations of PR extract). Specifically, both the control and the LPS group were administered 1 mL of saline i.g. for 21 days. Following this period, the control group received an intratracheal instillation of 100 μL of saline, while the LPS group received an intratracheal instillation of 100 μL of LPS (6 mg·kg^−1^). The lung tissue was collected from each group 24 h later. In the three PR-A967079-LPS and three PR-AITC-LPS groups, the mice were i.g. administered the PR extract daily for 21 days, with doses of 0.8, 1.6, and 3.2 mg·kg^−1^ for each group. After this period, the mice in these groups were administered either A967079 (60 mg·kg^−1^) or AITC (5 mg·kg^−1^) intraperitoneally, respectively. After 1 h, all the groups received an intratracheal instillation of 100 μL of LPS (6 mg·kg^−1^). The lung tissue was collected from each group 24 h later. Lastly, in the PR groups, the healthy mice were i.g. administered the PR extract daily for 21 days, with doses of 0.8, 1.6, and 3.2 mg·kg^−1^ for each group. After this period, the lung tissue was collected.

### 4.3. Cell Culture 

BEAS-2B cells were obtained from the Beijing Beina Chuanglian Biotechnology Research Institute and cultured in DMEM fortified with 10% FBS, along with penicillin (100 μg/mL) and streptomycin (100 μg/mL). The incubation was performed at a temperature of 37 °C under an atmosphere containing 5% CO_2_.

### 4.4. Histopathological Examination of Lung Tissue

Histopathological analysis was performed as detailed in a previously reported article [31]. In brief, paraformaldehyde-fixed fresh lung tissue from the mice was embedded in paraffin. The wax blocks were cut into 4 µm slices and baked. The tissue sections were subsequently stained with hematoxylin and eosin (H&E). At the same time, the lung histomorphology was independently evaluated by two pathologists. The lung injury was scored based on four parameters, namely, alveolar septal thickness (A), alveolar hemorrhage (B), intra-alveolar fibrin deposition (C), and intra-alveolar inflammatory cell infiltration (D), each scored from 0 to 3. The mean score for each experimental group was then computed.

### 4.5. Analysis of Tumor Necrosis Factor-α (TNF-α) and Interleukin-1β (IL-1β) in the Bronchoalveolar Lavage Fluid

Bronchoalveolar lavage fluid (BALF) was obtained from mice that had been euthanized, following a methodology previously described [32]. Following is a brief outline of the procedure. A sterile intravenous catheter was inserted into the trachea, and the BALF was collected by flushing with 1 mL of saline solution three times. Following this, the BALF samples were centrifuged at a speed of 1000 rpm (or, equivalently, 134× *g*) for a duration of 5 min. The collected supernatants were then stored at a temperature of −80 °C until further analysis. The levels of TNF-α and IL-1β in these samples were quantified using readily available ELISA kits, strictly adhering to the guidelines provided by the kit manufacturer.

### 4.6. Western Blotting

The analysis of TRPA1 in lung tissue was performed using Western blotting, as per the procedures elucidated in our prior work [33]. Briefly, the lung tissue was freshly extracted and lysed in an ice-cold buffer for a period of 10 min. The BCA Protein Assay Kit was utilized to quantify the proteins. The proteins were then separated using a 10% SDS polyacrylamide gel electrophoresis (SDS-PAGE) procedure and subsequently transferred onto polyvinylidene fluoride (PVDF) membranes. These membranes were blocked using a solution of 5% nonfat dry milk in 1× TBST and kept at room temperature for 2 h. Following a thorough washing, the membranes were left to incubate with the primary antibody at 4 °C overnight. The membranes were then washed three times with 1× TBST before incubating them with the horseradish peroxidase-conjugated secondary antibody for another 2 h at room temperature. Protein detection was achieved through the use of an Enhanced Chemiluminescence Kit (ECL), and quantification of the immunoblots was accomplished using Image J software.

### 4.7. Assay of the mRNA Expression of TRPA1 by qPCR

The extraction of total RNA from the lung tissue of the mice was achieved utilizing the TRIZOL reagent (Invitrogen, Carlsbad, CA, USA). RNA quality was examined by concentration detection, in which the absorbance ratio at OD260/OD280 was 1.8–2.0, and that at OD260/OD230 was 2.0–2.2 [34]. RNA integrity was determined through 18S and 28S rRNA analysis by gel electrophoresis (Appendix A). The reverse transcription process was conducted using the PrimeScript™ RT reagent Kit (Japan, Takara), in strict adherence to the protocol specified by the manufacturer. Following this, the synthesized cDNA was subjected to real-time PCR using the Light Cycle 480 system (Roche, Basel, Switzerland) [35]. The housekeeping gene GAPDH was employed as an internal reference. The resulting data were analyzed via the 2^−ΔΔCt^ method of relative quantification [36]. The TRPA1 and GAPDH gene primers were designed and synthesized by Beijing Rui Bo Xing Ke Biotechnology Co (Beijing, China). [37]. The primer sequences are as follows (Table 2):

### 4.8. Composition Analysis of the PR Water Extract with UPLC-Q-TOF-MS/MS

The water extract of PR was subjected to an analysis with the UPLC-Q-TOF-MS/MS technique (utilizing the Waters Xevo G2-XS UPLC-Q/TOF) to pinpoint potential bioactive constituents. This process involved chromatography performed on an Agilent SB-C18 column (2.1 × 100 mm i.d., with particle size of 1.8 μm); the temperature of the column was consistently held at 35 °C. The flow rate for the process was maintained at 0.4 mL per minute and the injected sample volume was 5 μL. The Q-TOF-MS/MS parameters included the analysis of the samples in negative ion modes with the mass-to-charge (*m*/*z*) scanning range set from 120 to 1600. The Ion Spray Voltage Floating (ISVF) was adjusted to 4500 V, and the ion source temperature (TEM) was fixed at 500 °C.

### 4.9. Molecular Docking

To simulate the interaction between TRPA1 and platycodins, molecular docking studies were conducted, examining hydrogen bonds, attractive van der Waals interactions, electrostatic interactions, and attractive hydrophobic interactions [38]. The relationship between the five platycodins and TRPA1 was assessed by calculating receptor–ligand free energy and binding affinity.

### 4.10. Immunofluorescence

Cellular immunofluorescence staining was conducted to detect TRPA1 expression in BEAS-2B cells. Briefly, BEAS-2B cells were seeded in 12-well plates (3 × 104 cells/well). After 24 h of simultaneous stimulation with different doses (3.5, 7 and 14 µM) of PD_3_ and 1.5 µM LPS, the cells were subjected to labeling with the TRPA1 antibody. Fluorescently labeled secondary antibody was then added, the cells were incubated at room temperature in the dark for 1 h, and subsequently the supernatant was discarded [39]. Finally, the cell nuclei were stained with DAPI for 1 min, and cellular fluorescence was monitored using immunofluorescence microscopy.

### 4.11. Measurement of Ca^2+^ Influx

The intracellular calcium levels were measured using the Fluo-3-AM calcium assay kit, following the instructions of the manufacturer. In brief, after 24 h of treatment with different does (3.5, 7, and 14 µM) of PD_3_ under 1.5 µM LPS stimulation, BEAS-2B cells were washed three times and stained with 5 μM Fluo-3-AM for 30 min at 37 °C in the dark. After washing, the fluorescence level was measured again.

### 4.12. Statistical Analysis

The graphs were obtained and the statistical analyses were performed using GraphPad Prism 8.0 software (GraphPad, San Diego, CA, USA). Data are presented as mean ± standard error of mean (SEM) of at least 3 or more independent experiments. ANOVA and Duncan’s multiple range tests were used for data significance analysis.

## Figures and Tables

**Figure 1 molecules-28-05213-f001:**
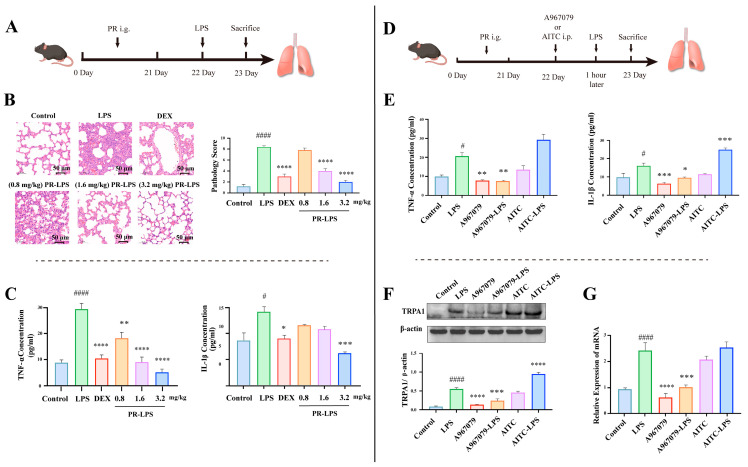
(**A**) Protective effects of PR on pneumonia in mice. (**B**) H&E staining of lung tissue and histologic scoring. (**C**) The concentrations of TNF-α and IL-1β were measured in BALF samples. (**D**) The mediating role of the TRPA1 receptor in pulmonary inflammation. (**E**) The TNF-α and IL-1β concentrations in the BALF were determined by ELISA. (**F**) TRPA1 protein expression was measured by Western blot analysis in lung tissue. (**G**) TRPA1 gene expression levels in lung tissue. All experiments were run in parallel (*n* = ≥3). Significant differences are indicated in the figures and legends as # *p* < 0.05, and #### *p* < 0.0001 vs. the Control; * *p* < 0.05, ** *p* < 0.01, *** *p* < 0.001 and **** *p* < 0.0001 vs. LPS-induced mice (LPS).

**Figure 2 molecules-28-05213-f002:**
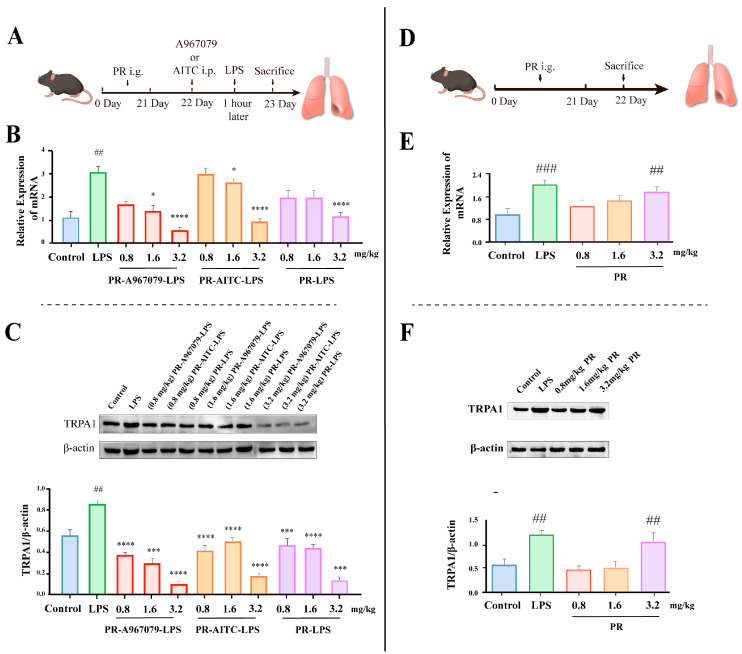
(**A**) PR inhibits TRPA1 expression to prevent LPS-induced pulmonary inflammation. (**B**) The expression levels of the TRPA1 gene in the lung tissue. TRPA1 (**C**) protein expression was measured by Western blot analysis in lung tissue. Since the number of mice in the experimental group amounted to 11, the high-dose group was analyzed in a second block of prefabricated gel. All experiments were run in parallel (*n* = ≥3). Significant differences are indicated in the figures and legends as ## *p* < 0.01, ### *p* < 0.001 vs. the Control; * *p* < 0.05, *** *p* < 0.001 and **** *p* < 0.0001 vs. LPS-induced mice (LPS). (**D**) Effect of PR on TRPA1 in mice. (**E**) The expression levels of the TRPA1 gene in the lung tissue. Experimental scheme (**F**) TRPA1 protein expression in lung tissue. Significant differences are indicated in the figures and legends as ## *p* < 0.01, ### *p* < 0.001 vs. the Control.

**Figure 3 molecules-28-05213-f003:**
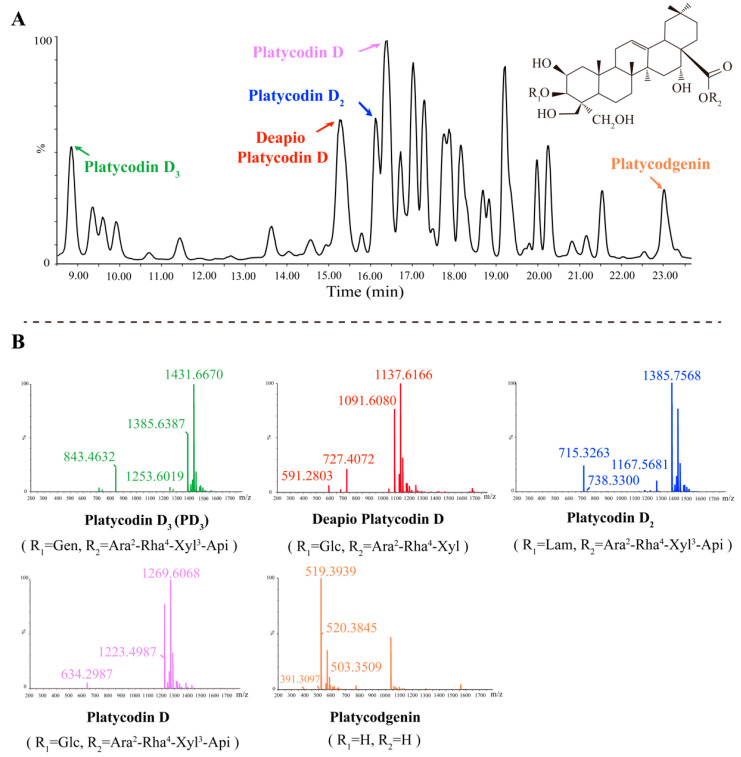
(**A**) The total ion chromatograms of PR by 2D UPLC/Xevo G2-XS-Q/TOF. (**B**) ESI-MS/MS spectrum of five platycodins.

**Figure 4 molecules-28-05213-f004:**
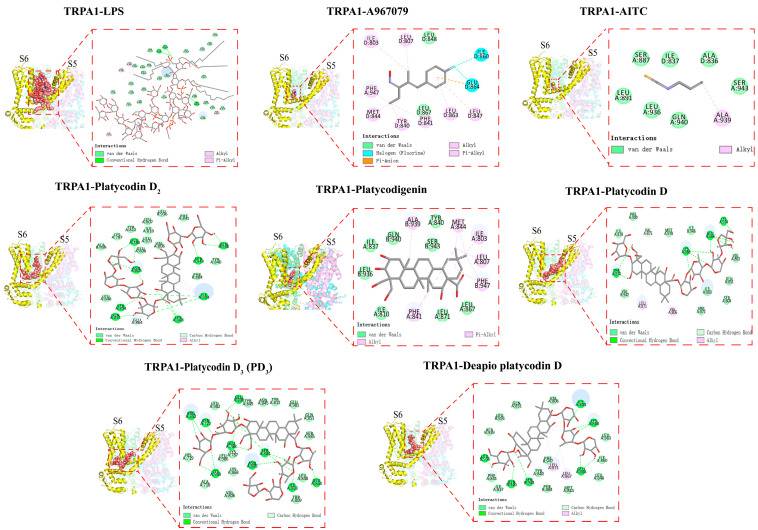
The binding mode of the TRPA1 protein with the LPS, A967079, AITC, and platycodins.

**Figure 5 molecules-28-05213-f005:**
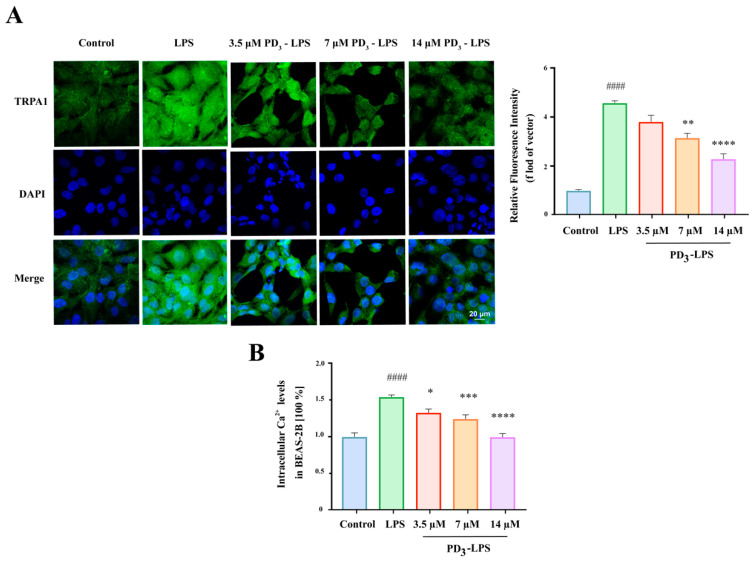
(**A**) Immunofluorescence assay to detect the TRPA1 protein in BEAS-2B cells, scale bar, 20 μm. Green fluorescence corresponds to TRPA1, nuclei are stained in blue. (**B**) Histogram showing TRPA1 quantification in BEAS-2B cells. (**C**) Effect of PR on intracellular Ca^2+^ levels in BEAS-2B cells. Significant differences are indicated in the figures and legends as #### *p* < 0.0001 vs. the Control; * *p* < 0.05, ** *p* < 0.01, *** *p* < 0.001 and **** *p* < 0.0001 vs. LPS-induced mice (LPS).

**Table 1 molecules-28-05213-t001:** Platycodigenin types identified in PR.

Components	Rt (Min)	Formula	R1	R2	Ion Fragmentation
Platycodin D_3_	8.768	C_63_H_102_O_33_	Gen	Ara^2^-Rha^4^-Xyl^3^-Api	1431.6670, 1385.6387, 1253.6019, 843.4632
Deapio-Platycodin D	15.36	C_52_H_84_O_24_	Glc	Ara^2^-Rha^4^-Xyl	1137.6166, 1091.6080, 727.4072, 591.2803
Platycodin D_2_	16.17	C_63_H_102_O_33_	Lam	Ara^2^-Rha^4^-Xyl^3^-Api	1385.7568, 1167.5681, 738.3300, 715.3263
Platycodin D	16.27	C_57_H_92_O_28_	Glc	Ara^2^-Rha^4^-Xyl^3^-Api	1269.6068, 1223.4987, 634.2987
Platycodigenin	22.96	C_32_H_48_O_7_	H	H	520.3845, 519.3939, 503.3509, 391.3097

The chemical structures of platycodins from PR. Glc, A, d-glucuronic acid; Ara, α-l-arabinopyranosyl; Rha, α-l-rhamnopyranosyl; Xyl, β-d-xylopyranosyl; Api, β-d-apiofuranosyl; Gen (gentiobiosyl), Glc^6^-Glc; Lam (laminaribiosyl), Glc^3^-Glc.

**Table 2 molecules-28-05213-t002:** Primers used for RT-qPCR analysis.

Gene	Sequence (5′-3′)	Product Length/bp	Annealing Temperature	Cycle Number
TRPA1	F: CGAGAGTCCTTCCTAGAACCAT R: CCTCAGCAATGTCGCCAAC	145	62 °C	40
GAPDH	F: TGCCCCCATGTTTGTGATG R: TGTGGTCATGAGCCCTTCC	126	62 °C	40

## Data Availability

Not applicable.

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
