# Peer review of "Platycodonis Radix Alleviates LPS-Induced Lung Inflammation through Modulation of TRPA1 Channels"

_molecules, 2023, doi:10.3390/molecules28135213_

Round 1

Reviewer 1 Report

The authors report studies investigating the 1effects of PR treatment on LPS-induced lung inflammation and propose that PR modulates TRPA1 channels.  This is an interesting report.

Specific Comments

1.The authors show results in figure 1 investigating effects of PR on lung histology and cytokine production. Presumably dexamethasone is a positive control, but there is no mention of the DEX results in the text and how they compare to PR treatment.  

2. Is the resolution the same in all of the panels of Figure 1B?  showing a scale bar would help.

3. For Figure 1C, the authors state that there is a dose-responsive reduction in TNF and IL-1B. The inhibition of TNF do look dose-responsive, whereas the IL-1B decrease only occurs at the highest concentration.

4. In Figure 1F, the authors presumably show changes in TRPA1 message; however, the blot is so dark that it is difficult to evaluate.  Are there better blots available?

5. The number of experiments performed should be indicated for Figure 1.

6. In Figure 2B and 2C, the red and orange bars are labeled the same. Are these mislabeled?  This is confusing.

7. The effects of A967079 and AITC shown in Figure 2 do not seem to be much different than PR alone.  However, the modeling data proposes that the PR and AITC sites on TRPA1 are similar. This should be clarified.

8. The molecular docking studies provide supporting data but do not demonstrate that platycodins actually bind to TRPA1.  This should be clarified in the text, since the abstract states that molecular docking confirmed binding interactions.

9. In figure 5, the authors show data on TRPA1 in BEAS-2B cells.  The data show PD3 treatment leads to decreased immunofluorescence and calcium flux in LPS-treated cells. What is the effect of the TRPA1 inhibitor in these studies?

10. The authors should explain how PR or PD3 cause reduced expression of TRPA1 is TRPA1 is also the target of PR/PD3.

English is fine.  There are a few typos or incorrect grammar uses.

Author Response

Dear Reviewers,

Thank you for reviewing our manuscript and for the constructive comments, which greatly helped us to improve the manuscript “Platycodonis Radix alleviates LPS-induced lung inflammation through modulation of TRPA1 channels” (ID: molecules-2468579). The manuscript was carefully revised, and point-by-point response was listed below. We hope that your comments have been addressed accurately. The revised manuscript was marked with yellow color and the responses were presented in red text.

Q1. The authors show results in figure 1 investigating effects of PR on lung histology and cytokine production. Presumably dexamethasone is a positive control, but there is no mention of the DEX results in the text and how they compare to PR treatment.  

A: Thank you for your valuable comment. We have added relevant DEX descriptions in the manuscript (Line 87 to 88).

Q2. Is the resolution the same in all of the panels of Figure 1B?  showing a scale bar would help.

A: Yes, all panels have the same resolution. We have added a clear scale in Figure 1B.

Q3. For Figure 1C, the authors state that there is a dose-responsive reduction in TNF-α and IL-1β. The inhibition of TNF do look dose-responsive, whereas the IL-1β decrease only occurs at the highest concentration.

A: Thank you for your comments. Upon observing IL-1β in the context of PR-LPS treatments (0.8, 1.6 mg/kg), we note a minor downward trend in expression levels. However, this difference is not statistically significant when compared to LPS alone.

Q4. In Figure 1F, the authors presumably show changes in TRPA1 message; however, the blot is so dark that it is difficult to evaluate.  Are there better blots available?

A: Thank you for your comment. We have updated the Western blot images in Figure 1F.

Q5. The number of experiments performed should be indicated for Figure 1.

A: Thank you for your suggestion. The number of experiments performed has been added in Figure 1 (Line 132).

Q6. In Figure 2B and 2C, the red and orange bars are labeled the same. Are these mislabeled?  This is confusing.

A: We are sorry for our careless mistake. The label have been changed in Figure 2.

Q7. The effects of A967079 and AITC shown in Figure 2 do not seem to be much different than PR alone.  However, the modeling data proposes that the PR and AITC sites on TRPA1 are similar. This should be clarified.

A: Thank you for your valuable suggestion. We have added more discuses in the manuscript (line 120-123).

Q8. The molecular docking studies provide supporting data but do not demonstrate that platycodins actually bind to TRPA1.  This should be clarified in the text, since the abstract states that molecular docking confirmed binding interactions.

A: Thank you for your kindness comments. The statement about molecular docking in the Abstract has been modified to more accurately reflect our purpose (Line 31-33).

Q9. In figure 5, the authors show data on TRPA1 in BEAS-2B cells.  The data show PD3 treatment leads to decreased immunofluorescence and calcium flux in LPS-treated cells. What is the effect of the TRPA1 inhibitor in these studies?

A: Thank you for your valuable input on the manuscript. In the context of our BEAS-2B cell experiment, we used PD3 to demonstrate its role as a channel regulator of TRPA1. When TRPA1 is inhibited by PD3, our data shows a decrease in immunofluorescence and calcium flux in LPS-treated cells. This suggests that TRPA1 inhibition via PD3 might play a crucial role in modulating these cellular responses.

  1. The authors should explain how PR or PD3cause reduced expression of TRPA1 is TRPA1 is also the target of PR/PD3.

A: Thank you for your insightful question. As for PD3, it is an active ingredient with a high affinity for TRPA1 and has demonstrated anti-inflammatory properties. Our in vitro results show that PD3 can effectively inhibit the activation of the TRPA1 channel and the influx of Ca2+, a key ion involved in both the activation and modulation of TRPA1 channel activity. By inhibiting this inward flow of Ca2+, PD3 inhibits TRPA1, potentially alleviating LPS-induced lung inflammation (Line 288-291).

We are looking forward to receiving your response, and please contact me without any hesitation if you have any question.

Best Regards,

Yours sincerely,

Junbo Xie

E-mail: xiejb@tjutcm.edu.cn..

Reviewer 2 Report

The authors performed a good work with bioactive platycodins.

Minor points:

1. Please provide the supporting information file (containing original images and HPLC standard platycodins).

2. Please provide more details about the characterization of platycodin derivatives: adding more references and giving a discussion about the main fragmentation of platycodins. Revise the molecular formula of compounds in L153-154.

3. Please provide bigger and better docking images of compounds in Figure 4. Currently, no H-bonds could be observed.

Author Response

Dear Reviewers,

Thank you for reviewing our manuscript and for the constructive comments, which greatly helped us to improve the manuscript “Platycodonis Radix alleviates LPS-induced lung inflammation through modulation of TRPA1 channels” (ID: molecules-2468579). The manuscript was carefully revised, and point-by-point response was listed below. We hope that your comments have been addressed accurately. The revised manuscript was marked with yellow color and the responses were presented in red text.

Q1. Please provide the supporting information file (containing original images and HPLC standard platycodins).

A: Thank you for your valuable suggestions. The associated resulting images have been added in supporting information.

Q2. Please provide more details about the characterization of platycodin derivatives: adding more references and giving a discussion about the main fragmentation of platycodins. Revise the molecular formula of compounds in L153-154.

A: Thank you for your valuable suggestions. We have added more discuss and added reference in the manuscript and modified the molecular formula. (Line 166-173)

Q3. Please provide bigger and better docking images of compounds in Figure 4. Currently, no H-bonds could be observed. 

A: Thank you for your valuable suggestions. We have modified the images to make them clearer (Figure 4).

We are looking forward to receiving your response, and please contact me without any hesitation if you have any question.

Best Regards,

Yours sincerely,

Junbo Xie

E-mail: xiejb@tjutcm.edu.cn.

Reviewer 3 Report

The manuscript needs revision. Please refer to comments given in the text of reviewed attached file of the manuscript.

Author Response

Dear Reviewers,

Thank you for reviewing our manuscript and for the constructive comments, which greatly helped us to improve the manuscript “Platycodonis Radix alleviates LPS-induced lung inflammation through modulation of TRPA1 channels” (ID: molecules-2468579). The manuscript was carefully revised, and point-by-point response was listed below. We hope that your comments have been addressed accurately. The revised manuscript was marked with yellow color and the responses were presented in red text.

Q1. please add full names for abbreviations in the first place of text!!

A: Thank you for your comments. We have revised all the terms in the manuscript (Line 19 to 20).

Q2. please add about M and M in the abstract before results.

A: Thank you again for providing us with valuable suggestions for the manuscript. We have revised the abstract as suggested (Line 23 to 35).

Q3. please add space before [ ].

A: Sorry for the careless mistake. We have revised it in the manuscript.

Q4. It is better explain about importance and application of plant products in the text. For this you can use below sentences and references:the use  of  plant  products  is  increasing  in  many  segments  of  the population (Jafari Ahmadabadi et al., 2023).  According to an estimate, 80% of the world's population relies upon plants for their medication (Heidarpour et al. 2011). Most of the synthetic drugs used at present for analgesic and anti-nociceptive effect have many side and toxic effects (Mohammadabadi et al., 2009). Plants still represent a large untapped source of structurally novel compounds that might serve as lead for the development of novel drugs. Many medicines of plant origin with analgesic and anti-nociceptive activity had been used since long time without any adverse effect (Mohammadabadi et al., 2009).

Heidarpour F, Mohammadabadi MR, ISM Zaidul, B Maherani, N Saari, AA Hamid, F Abas, MYA Manap, MR Mozafari 2011. Use of prebiotics in oral delivery of bioactive compounds: a nanotechnology perspective. Pharmazie 66 (5), 319-324

Jafari Ahmadabadi SAA, Askari-Hemmat H, Mohammadabadi M, Asadi Fozi M, Mansouri Babhootki M (2023) The effect of Cannabis seed on DLK1 gene expression in heart tissue of Kermani lambs. Agricultural Biotechnology Journal 15 (1), 217-234.

Mohammadabadi MR, El-Tamimy M, R Gianello, MR Mozafari 2009. Supramolecular assemblies of zwitterionic nanoliposome-polynucleotide complexes as gene transfer vectors: Nanolipoplex formulation and in vitro characterisation Journal of liposome research 19 (2), 105-115.

A: Thank you for your valuable suggestions. We have added more discusses and references to support this idea (Jafari Ahmadabadi et al., 2023; Heidarpour et al. 2011; Mohammadabadi et al., 2009) (Line 54 to 58).

Q5. In the text of the introduction, it is not clear what the problem is? And what problem do you want to solve? Please add your aim of this study at the end of introduction section. Please specify in the objective whether your research is being conducted for the first time in the world or is it a continuation of another research? What is the superiority of your research compared to other researches?

A: Thank you for your valuable comments. We have updated the introduction and discussion sections in the manuscript (Line 65 to 67 & Line 74 to 75).

Q6. please add a figure from extracted RNA on the agarose gel.

A: Thank you for your valuable suggestions. We have put the result of extracted RNA on the agarose gel on supporting information.

Q7. If you can sequence the cDNA and compare it with the existing sequences, the results will be much more complete, and you can make more accurate and reliable conclusions.

A: We appreciate your insightful suggestion. Indeed, sequencing the cDNA and comparing it with existing sequences could provide a deeper understanding of the genetic alterations that might be associated with TRPA1's function. Such an approach could certainly enhance the robustness of our conclusions. While this was not incorporated into our current study, we recognize its importance and are planning to include cDNA sequencing in our subsequent investigations to further clarify the mechanisms of action of TRPA1.

Q8. It is better to compare your results with results of other researchers and mention that their confirm or reject your results more.

A: Thank you for valuable comments. The more discuss was added in the manuscript (Line 277 to 282).

Q9. Did you check quantity and quality of extracted RNA? Please explain in the text of the manuscript. please explain how did you it or add reference for this. Moreover, please add country for used kits and devices. please add country for used kits and devices. did you design these primers or derived from other references?

If yes, please explain how? and add used software for designing with its reference.

If no, please add reference from which you selected these primers

it is better writing these primers in a table and add below information for every primer:

accession number、PCR product length、annealing、temperature

A: Thank you for valuable suggestions and comments. Relevant details have been added in the manuscript (Line 396 to 399, Line 404 to 406 and Table 2).

We are looking forward to receiving your response, and please contact me without any hesitation if you have any question.

Best Regards,

Yours sincerely,

Junbo Xie

E-mail: xiejb@tjutcm.edu.cn.

Round 2

Reviewer 1 Report

The authors have addressed my comments. and have improved the manuscript.